# An estimate of absolute shear-wave speed in the Earth's inner core

Thuany Costa de Lima [1] ✉, Thanh-Son Phạm [1], Xiaolong Ma [1] & Hrvoje Tkalčić [1]

Observations of seismic body waves that traverse the Earth's inner core (IC) as shear (J) waves are critical for understanding the IC shear properties, advancing our knowledge of the Earth's internal structure and evolution. Here, we present several seismological observations of J phases detected in the earthquake late-coda correlation wavefield at periods of 15–50 s, notably via the correlation feature I-J, found to be independent of the Earth reference velocity model. Because I-J is unaffected by compressional wave speeds of the Earth's inner core, outer core, and mantle, it represents an autonomous class of seismological measurements to benchmark the inner core properties. We estimate the absolute shear-wave speed in the IC to be 3.39 ± 0.02 km/s near the top and 3.54 ± 0.02 km/s in the center, lower than recently reported values. This is a 3.4 ± 0.5% reduction from the Preliminary Reference Earth Model (PREM), suggesting a less rigid IC than previously estimated from the normal mode data. Such a low shear-wave speed requires re-evaluating IC composition, including the abundance of light elements, the atomic properties and stable crystallographic phase of iron, and the IC solidification process.

The speed of shear waves propagating through the Earth's inner core (IC), also known as J waves, is an essential parameter for characterizing the composition[1], physical state[2–5], crystallographic structures[6–8], and the abundance of light elements[9–11] in the iron alloys constituting the IC at high pressures and temperatures. This understanding, in turn, sheds light onto the IC's solidification process[12], thermal history[13], and dynamics[12] over time. In the past, not long after the IC's discovery[14], the IC was hypothesized to be in a solid state mainly composed of crystalized iron[15], although there are alternative views[16]. Early support for the solidity of the IC came through observation of normal modes[17,18], characterized by long-period standing waves oscillating in the entire Earth's interior several hours to days after large earthquakes. Some observations of normal modes could not be explained otherwise without the presence of a solid IC, and these observations were subsequently used to construct the 1D profile of shear-wave speeds in the Preliminary Reference Earth Model (PREM)[19].

Since the normal-mode studies suggested the solidity of the IC, observations of body shear waves propagating through the IC were sought after. This is because such observations could provide the most direct evidence for the solidity of the IC (Fig. 1a), given that shear waves do not propagate through liquids. However, the search has proved challenging[20–25]. The seismic waves that leave a seismic source as either compressional (denoted as P) or shear (S) waves transverse the outer core as compressional (K) waves and then convert into shear (J) waves within the IC before ascending through the outer core and mantle to the surface of the Earth, such as PKJKP or SKJKP waves. Unlike the commonly observed IC compressional waves PKIKP, the theoretical amplitudes of PKJKP waves are so weak that they are under the observational threshold. This is mainly due to the weak compressional-shear energy conversions at the inner-outer core boundary (ICB) and the strong attenuation of shear waves within the IC[26]. These effects make detecting J waves in the direct seismic wavefield elusive[27], which is proved by synthetic experiments without the background seismic noise[27,28].

The recent advent of the global earthquake-coda correlation studies[29–33] has promised a new research avenue to shed more light on the shear properties of the IC. The Earth's coda-correlation wavefield[29,30] is an abstract wavefield manifesting the similarity of

[1]Research School of Earth Sciences, The Australian National University, Canberra, ACT, Australia. ✉e-mail: Thuany.CostadeLima@anu.edu.au

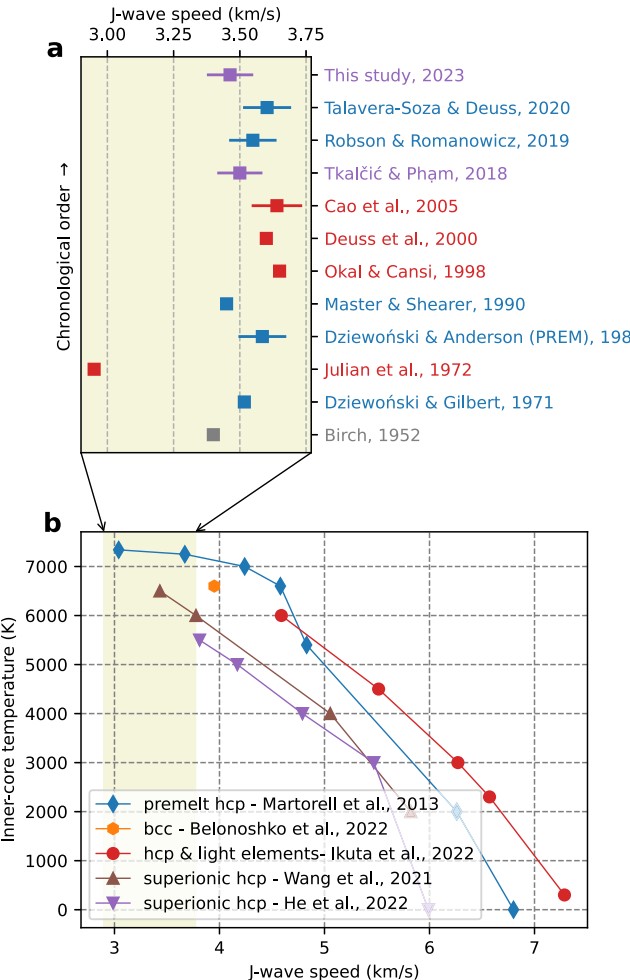

**Fig. 1 | Compilation of J-wave speed estimates from seismological observations and recent mineral physics simulations. a** Different values of J-wave speeds obtained from multiple IC-solidity studies. Colors represent different methodologies (blue: normal modes, red: shear body waves, purple: coda-correlation wavefield, and gray: mineral physics estimates). Squares denote the averaged J-wave speeds, and the bars represent the range of J-wave speeds from the inner-core boundary (ICB) to the Earth's center, available in some studies. **b** Summary of recent mineral physics hypotheses[2,3,5,37,38] providing possible explanations for seismologically observed low J-wave speeds. Colored lines show different J-wave speeds as functions of temperatures at 360 GPa, the pressure at the Earth's center[4]. The shaded area corresponds to the range of J-wave speeds shown in panel (**a**).

weak seismic waves propagating through the deep Earth's interior in the late parts of the seismogram, known as the coda of large earthquakes. The global coda-correlogram (Fig. 2a, b), a 2D image of inter-receiver distance and lapse time, is a realization of the correlation wavefield. It is constructed by cross-correlating hours of records of large earthquakes at hundreds to thousands of seismic stations (see Methods section).

This study reports several correlation features highly sensitive to J-wave speeds (Fig. S1), most previously undetected but now becoming observable thanks to recently improved methods in constructing global correlograms[34,35]. We then focus on a unique correlation feature named I-J, previously noticed in synthetic simulations but undetected in the observed coda-correlation data[36]. Interestingly, we find that this feature's timing is minimally sensitive to the background velocity models used for other parts of the Earth (see the Methods section). By fitting the observation with synthetic correlation wavefield, our estimates of J-wave speeds near the inner-core boundary (ICB) and in the center of the Earth are 3.39 ± 0.02 km/s and 3.54 ± 0.02 km/s,

respectively, equivalent to 3.4% reduction from the PREM[19] values. We discuss these estimates in the context of several hypotheses recently proposed by mineral physicists to explain the low J-wave speed detected in seismological observations (Fig. 1b). These include the presence of partial melt pockets[4], nonlinear effects of J wave near the melting temperature[3] (i.e., premelt), the superionic IC and the abundance of light-elements[2,5,37], and stability of *bcc* iron crystallographic structures[38].

## Results

### Enhancement of global correlogram and identification of J-wave correlation features

Until recently, the common practice of constructing the global correlogram was based on cross-correlating records of most available earthquakes, without consideration for the source mechanism nor the geometry of receiver pairs, both of which were eventually proven to have a detrimental effect on the quality of the correlogram[35,39–41]. In this study, we enhance the prominence of correlation features (Fig. 2a) by accounting for the advances in understanding their formation mechanism[35,39–41] (see Methods section). Firstly, we select, cross-correlate, and stack only records of earthquakes showing normal and thrust faulting mechanisms (Fig. 2c) at station pairs with the difference between great-circle planes smaller than 20°. Earthquakes of strike-slip and oblique fault mechanisms are rejected because they do not radiate seismic energy favorably to the Earth's center[41]. Secondly, we discard visually poor coda-correlograms resulting from a small number of seismic records or complex source time functions (Fig. 2d). The combination of source mechanism and receiver pair geometry selections reduces the number of cross-correlated waveforms in the global stack. It makes the correlogram visually snowy (Fig. 2b) but greatly increases the overall expression of all correlation features and facilitates their identification. As demonstrated later, this selection allows for unambiguous detections of several J-wave features and a refinement of the previous estimate of the shear-wave speeds in the Earth's IC[36].

Correlation features sensitive to shear properties in the IC are identified systematically in a series of numerical experiments. Synthetic coda correlograms are calculated for a group of Earth's models, which use PREM[19] as the background model but with shear wave speeds in the IC perturbed from their original values. Because the IC is in the Earth's center and accounts for less than 1% of its volume, most features in the global correlograms remain unchanged. In contrast, correlation features sensitive to J-wave speeds can be visually detected when their timing in the vertical axis changes, corresponding to the speed variation in the IC, as shown in Supplementary Movie 1.

As a result of the exhaustive search, we identify four correlation features exhibiting outstanding sensitivity to the J-wave speeds in the synthetic correlograms (Fig. S1). These include I-J (observed between ~140° and 160° interstation distances in Figs. 3, 4, and S1), I2-J (a correlation feature in the shape of a cusp, with both positive and negative slowness legs observed at small interstation distances, Fig. S1), and two other correlation features, later referred to as x1-J and x2-J (Fig. S1). Their phase names indicate that one of the constituents of the correlation features propagates through the bulk IC as a shear wave (J), so they are sensitive to J-wave speeds. Schematic ray paths and possible constituents contributing to two unambiguously identified features, I2-J and I-J, are shown in Fig. 3. This figure sketches one family of seismic phase pairs (out of numerous other possibilities), xPKIKP{1,2}PKIKPn−xPKJKPPKIKPn, which share the large majority of propagation legs that contribute to the formation of the I-J and I2-J correlation features. The correlation features x1-J and x2-J are observed at larger times in the correlogram (Fig. S1). The non-uniqueness in determining their compressional-wave constituents (x1 and x2) and corresponding ray paths deter us from utilizing them (yet).

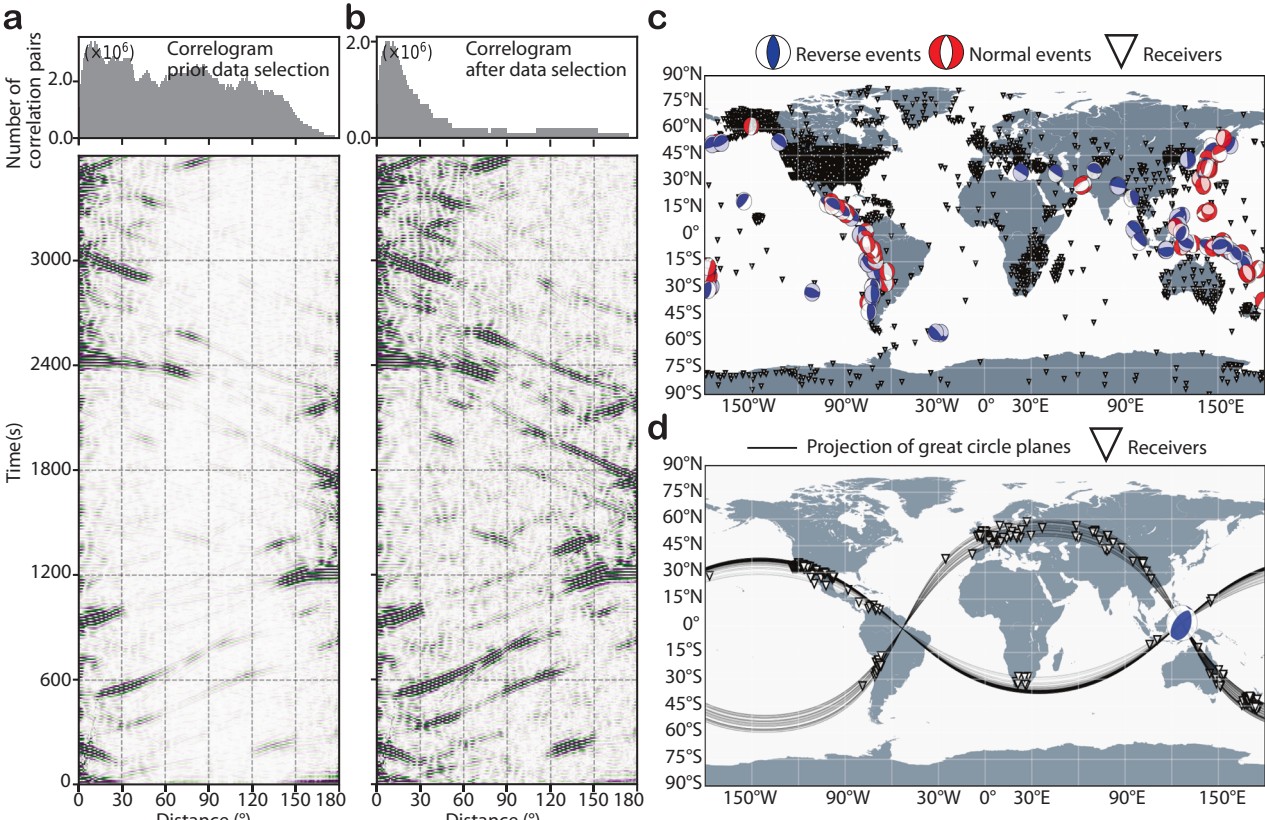

**Fig. 2 | Improvement of correlogram quality based on the selections of earthquake mechanisms and receiver pair geometries. a** Correlogram calculated from records of events of Mw ≥ 6.8 listed from 2010 to 2019 in the GCMT catalog, using all station pairs available in IRIS DMC (see Data Availability). **b** Correlogram after selecting normal and reverse events and station pairs that satisfy the back-azimuth condition (see Methods for details). **c** Map of 139 normal and reverse faulting earthquakes (beachballs in red and blue, respectively) and seismometer locations (triangles) used in this study. **d** Example of source-receiver pairs geometry analyzed in this study (see Methods for more details). Thin black lines represent the projection of the great circle plane between an earthquake in the Banda Sea region (blue beach ball) and stations (triangles) at azimuths 145 ± 10° and −123 ± 10°.

## Detection of J-wave correlation features in the observed data and estimate of absolute J-wave speed

Although J-wave correlation features can be seen in the synthetic correlogram (Fig. 4a, Supplementary Movie 1), their observational counterparts are not so easily recognized visually in the observed correlogram (Fig. 4b, Fig. S2). Figure 4b shows the observed waveforms of the I-J correlation feature compared to its synthetic counterpart. Despite its small amplitude, I-J can be observed clearly through the slant stack with phase-weighted stacking of the waveforms (see Methods section). When the slant-stacks are calculated for a range of ray parameters, the timing and slowness of observed correlation features can be identified similarly to their synthetic counterpart (Fig. 4c−e for the I-J feature; Fig. S1b−f for the other three correlation features).

Figure 5 features a parameter search of shear waves in the IC to match the synthesized correlation waveforms to the observed counterparts. The optimal velocity obtained for the I-J fitting is a 3.4 ± 0.5% reduction in the IC shear-wave speed from PREM (Fig. 5a), which is stable for six different ray parameters from 1.5 to 2.0 s/°, with a step of 0.1 s/° (Fig. 5b). Similarly, the best fitting velocities obtained for the cusp of I2-J, negative and positive slowness legs are, respectively, 2.5 ± 0.5% and 2.8 ± 0.5% reduction from PREM J-wave speed (Fig. S3a, S3b). The optimal velocities for the two hitherto unexplored features, x1-J and x2-J, are a 4.4 ± 0.5% reduction from PREM J-wave speed (Fig. S3c, S3d). The shear-wave velocity estimates from the grid search relative to PREM and CCREM[42] (a model derived using coda-correlation wavefield) are reported in Table S1. The estimates based on all four features

vary between 2.5 ± 0.5% and 4.4% ± 0.5% reduction, but these values can be reconciled, and the cause for this variation is explained below.

To understand the range of estimates of J-wave speeds corresponding to different correlation features, we diagnose their sensitivities to the background compressional velocity model in all major layers of the Earth: the mantle, outer core, and inner core. The compressional wave speeds in each layer are either lowered by 1%, kept unchanged, or increased by 1% relative to the PREM values. We observe significant changes in the timing of x1-J, x2-J, and the positive and negative slowness legs of I2-J (Fig. S3), but the I-J feature remains unchanged despite the background model variations (Fig. 5c−e). These results explain the observed range of J-wave speeds. Because the correlation features x1-J, x2-J, and I2-J have different sensitivities to the background Earth velocity model, their waveform fit to the synthetic wavefield could be used in future studies to constrain seismic properties in the mantle, outer core, and inner core.

Given the feature I-J's unique and striking insensitivity to Earth background models, we argue that the 3.4% reduction from PREM values[19] inferred from this feature is an estimate of absolute J wave speed in a radially symmetric Earth's model. The reduction is more considerable than a 2.5% reduction[36] measured upon the I2-J's negative slowness leg, assuming PREM as a background Earth model. Note that we also obtain the same reduction of ~2.5% (Table S1) for the I2-J feature (both positive and negative slowness legs of the I2-J cusp) using not only different sets of earthquakes and stations (see Methods section) but also a different synthetic waveform generator (i.e., Yspec[43] instead of AxiSEM[44]; see Methods section), which ensures the

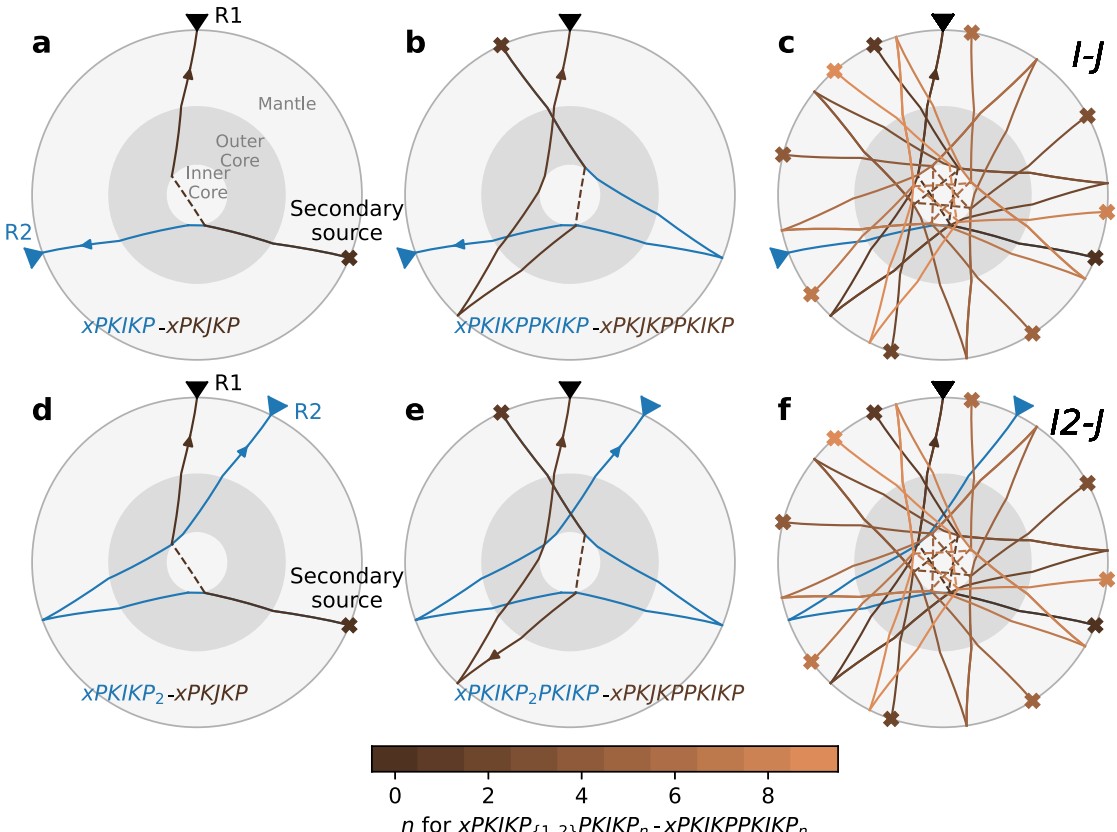

**Fig. 3 | Schematic formation mechanism of I-J and I2-J correlation features.**
**a** Ray paths of two possible constituents contributing to the formation of the I-J correlation feature, xPKIKP-xPKJKP. These constituents originate from a secondary source (i.e., via the Huygens principle), denoted by the x sign, and propagate as PKJKP and PKIKP waves to receivers R1 and R2. These two rays have the same ray parameter and share first propagation legs in the mantle and outer core. The secondary source x can be a surface reflection of any seismic ray propagating from a distant earthquake with the same ray parameter. The prefix x, whose travel time is canceled out and not sensitive to the I-J feature's timing, is used in the feature name to generically refer to the arbitrary ray paths before the two final constituents. **b** Similar to (a), but for a correlation feature with one more PKIKP leg. xPKIKPPKIKP-xPKJKPPKIKP is another example of constituent pairs whose differential travel times contribute to forming the I-J feature. **c** Similar to (a) and (b) but for many more PKIKP legs before arriving two the two receivers, i.e., xPKIKPPKIKP$_n$-xPKJKPPKIKP$_n$. The last PKIKP legs in common, $n = 0–10$, are color-coded as in the color bar. **d–f** Similar to (**a–c**) panels but for various constituent pairs contributing to forming I2-J correlation features.

measurements' robustness. Instead, the dependency on Earth background models caused differences in the estimates. Indeed, the values relying on the I2-J feature would be a 4.5% reduction if, for example, CCREM[42] were used (Table S1).

## Discussion

We consider the insensitivity of the I-J feature's timing to the Earth's background model as the most appealing property of the present estimate of shear-wave speeds in the IC. Although widely used in long-period seismic studies of the Earth's deep interiors, PREM[19] was constructed more than 40 years ago using primarily normal mode data. Recent refinements to PREM have been proposed[45], and new independent 1D reference models can be constructed as larger datasets of body-wave measurements[46,47] or global coda-correlation wavefield[42] become available. Multiple 1D Earth models providing good fitting for different global datasets highlight the inevitable uncertainties in these models in representing the Earth's radially averaged structures. Note that the mantle and outer core account for more than 99% of the Earth's volume, while the inner core is buried beneath them and accounts for less than 1%. Thus, the influence of the whole background models on the estimates of IC properties must be carefully considered. Hence, our assessment of J-wave speeds, minimally affected by the Earth velocity models and their uncertainties, serves as an independent benchmark for interdisciplinary research.

The 3.4% reduction from the PREM's J-wave values might enlarge the disagreement with recent normal mode studies[48,49] (Fig. 1a), although it is within the range constrained by PKiKP/P amplitude ratio measurements[50]. The apparent discrepancy between coda-correlation and normal mode estimates of J-wave speed could be explained by (i) strong dispersion of shear waves in the IC, (ii) sampling sensitivity, (iii) sensitivity to the background Earth's velocity models or combined effects of all three above factors. Firstly, the dispersion, characterizing velocities' dependence on frequencies, could play an essential role as the timing of correlation features is measured at 15–50 s, while normal mode data are at hundred-second-long periods. Secondly, normal modes' sensitivity vanishes near the inner core's center[48,51], where correlation features such as I2-J and I-J retain their sensitivities to the IC's very center (Fig. 3). Lastly, the inconsistency between normal modes and coda-correlation results for IC shear wave speed could be due to their different sensitivities to the Earth's background models, in a similar way the modeling of some correlation features is sensitive to the reference velocity models.

J-wave speeds in PREM are about 30% lower than ab initio calculations of pure iron[37,38] in mineral physics. The deficit is enlarged by this study's estimate of a further 3.4% reduction from PREM values. Yet, recent studies concerning preferred crystallographic iron structures in the IC conditions, hexagonal-close-packed (hcp) and body-centered-cubic (bcc), provide possible reconciliation for the deficit. Early hypotheses explaining the IC's low shear wave speed and high

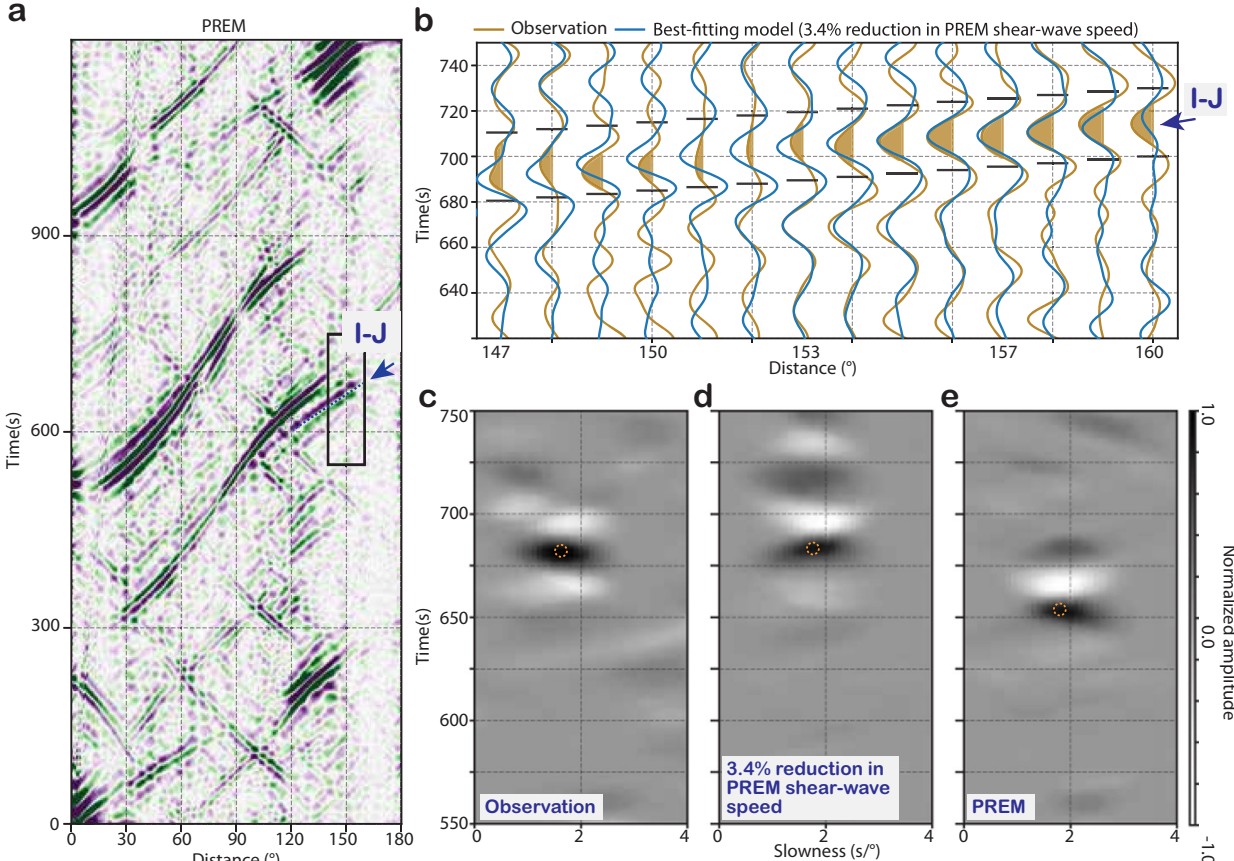

**Fig. 4 | Detection of I-J in the synthetic and observed correlation wavefield.**
**a** Synthetic correlogram estimated using PREM[19]. The inset marked by the black rectangle indicates the time and inter-receiver distance window predicted for I-J (highlighted by the dashed blue line near the blue arrow) used to calculate the slant stacks shown in (**c**) to (**e**). Negative amplitudes in the correlogram are in purple, and positive amplitudes are in green. **b** Waveforms of feature I-J (waveforms in dark yellow) extracted from the observed correlogram overlayed on the synthetics of the best-fitting model (waveforms in blue) calculated with an IC shear speed

reduction of 3.4% relative to PREM values. Negative amplitudes of I-J are filled in dark yellow to facilitate the visual inspection of their arrival times. Note the prominence of the signals (in between the horizontal solid black lines at each inter-station distance) in both observed and synthetic correlation waveforms. **c**–**e** Phase-weighted stacks of I-J detected in the observation (**c**), best-fitting model (**d**), and PREM (**e**). The orange circles on the slant stacks from (**c**) to (**e**) guide the comparison of features' timing.

attenuation involving *hcp* iron included the presence of partial melt pockets[4], and the non-linear properties of iron alloys near melting temperatures[3]. Singh et al.[4] suggested that partial melt pockets of about 10% in volume could explain the PREM's low J-wave speeds, so an even higher melt fraction would be required to explain our present estimate. The fractional melt volume could constrain the supercooling temperature, so the supercooling at IC's center would be lower than the estimated 100 K[13]. More recently, a significant non-linear drop in shear-wave speeds near the *hcp* iron's melting temperatures ranging about 7000–7340 K, known as the premelting effects, was suggested[3]. The temperature range is markedly higher than recent estimates of melting points for both possible candidates of crystallographic iron structures[52] (*hcp*: 6357 ± 54 K and *bcc*: 6168 ± 80 K at the ICB). Thus, the premelting non-linear drop of J-wave speeds[3] might not be a sustainable hypothesis[38].

The abundance of light elements in the IC is another popular hypothesis for explaining the known J-wave speed deficit[9]. Most recently, compressional and shear-wave speeds of *hcp* iron at the IC condition were measured[37] using a newly designed diamond anvil cell and inelastic X-ray scattering combined with X-ray diffraction. Their measurements for pure *hcp* iron are 4% higher than P-wave speeds and 29% higher than PREM's S-wave speeds. However, shear wave speeds as low as 2.95 km/s can be achieved by adding silicon and sulfur to the IC's

iron alloy (Fig. 1b). Independently, two recent studies[2,5] proposed that light elements can be highly diffusive between solid-state *hcp*-iron lattice. The coexistence of solid iron and highly mobile light elements is known as the superionic state, which exhibits significantly low shear-wave speeds and other observed seismological properties of the IC[36] (Fig. 1b).

Alternative from *hcp* as a candidate for stable crystallographic iron structures in the IC, *bcc* iron can also be stable in the IC condition[53,54]. Both *hcp* and *bcc* can explain distinct anisotropic properties of the innermost inner core, which was found seismologically to have a slow axis about 50–55° obliqued from the equator[55–60]. Additionally, the intrinsic properties of the *bcc* iron can explain the shear wave speed of 3.95 km/s[38], which is just 3.5% higher than PREM values (Fig. 1b). Thus, if the IC is mainly constituted by *bcc* iron, much less nickel and other light elements will be required to explain the range of seismologically observed J-wave speeds of the IC[38].

It is noteworthy that our study assumes a uniform variation in the bulk IC relative to the PREM-like radial distribution of shear-wave speeds. Thanks to that simplified assumption, the heterogeneous distribution of earthquakes and stations on Earth's surface contributing to the observed correlogram does not affect our estimated J-wave speeds. However, like the P-wave structure[51], the IC might exhibit a similar, if not higher, level of complexity in its S-wave

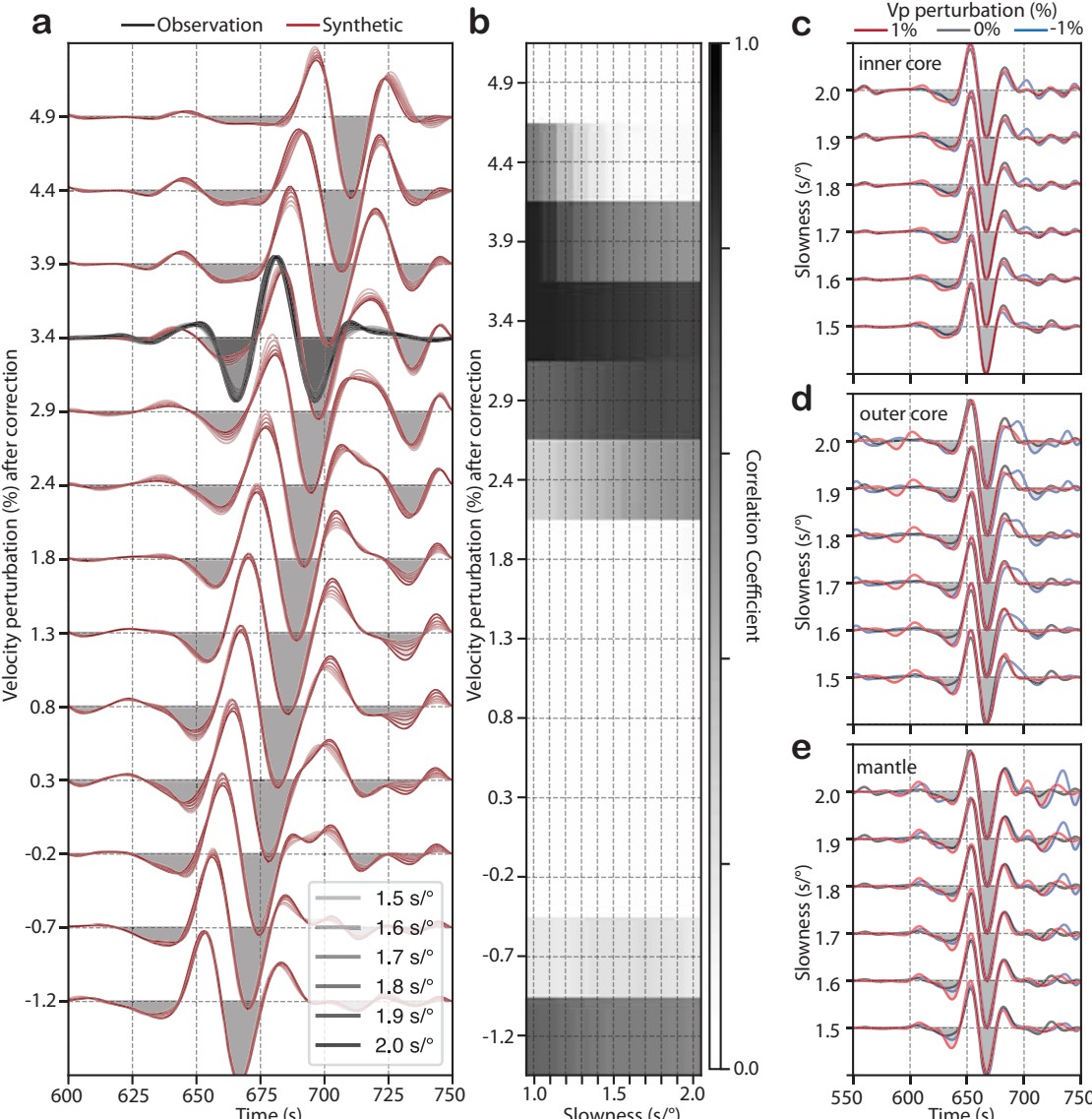

**Fig. 5 | An absolute estimation of shear-wave speed in the Earth's inner core. a** Waveform modeling of I-J. The observed waveforms are in black, and the synthetic waveforms are in red. The waveforms are phase-weighted stacks at the slowness of 1.5, 1.6, 1.7, 1.8, 1.9, and 2.0 s/° calculated using PREM as the reference model with J-wave speed perturbations from 0 to 6% reduction at the central period of $T = 23.1$ s which are corrected for the dispersion and displayed in the vertical axis at $T = 1$ s. The best-fitting model is chosen based on the coherence coefficient (CC; see Methods) with a 3.4% reduction in J-wave speed from PREM at $T = 1$ s. **b** Correlation coefficient matrix between observed and synthetic slant stacks. Note the consistency of the best-fitting model for the range of slowness from 1 s/° to 2 s/°. The vertical axis is the same as in (**a**). **c-e** Slant stacks of synthetic I-J estimated with perturbation in compressional velocity in (**c**) the inner core, (**d**) outer core, (**e**) and mantle, of +1%, 0%, and −1% relative to PREM, shown in blue, gray, and red, respectively. The waveforms are sorted as a function of slowness (vertical axis). Note that the arrival times of the I-J correlation feature are minimally influenced by variations in the Earth reference model, differently from other correlation features which provide a relative estimate of J-wave speed depending on the choice of the background velocity model (Figs. S3, 4).

structure[28]. Indeed, there has been initial evidence for IC shear-wave anisotropy[24,34], and shear-wave attenuation remains poorly understood[28]. Thus, we envision addressing the IC shear-wave structure complexity as an exciting future research topic. This might require a refined selection of event-receiver pairs and an even better understanding of the sensitivity kernel of the correlation wavefield to the Earth's structures, especially its finite frequency sensitivity for different period bands.

In conclusion, this study presents the intensified observational power and the correlation feature I-J with a direct sensitivity to the shear-wave speed in the IC minimally affected by the background Earth velocity model. We measure a 3.4% decrease from the PREM values throughout the IC's bulk, which is the latest, globally-averaged estimate of J-wave speed by the global correlation wavefield. Such low

J-wave speeds can be reconciled with the most recent advances in high-pressure mineral physics at the IC conditions. More generally, our results demonstrate a further advantage of the correlation wavefield in studying Earth's and other planets' interiors.

## Methods

### Data preparation and computation of global correlograms

We construct the correlogram using vertical component records of earthquakes of magnitude Mw ≥ 6.8 between 2010 and 2019, listed in the global centroid moment tensor catalog[61] (accessible at https://www.globalcmt.org/CMTsearch.html). The seismograms are downloaded from the Incorporated Research Institutions for Seismology Data Management Centre (IRIS-DMC) in the coda time window of 3–10 h after the event origin times. This coda time window is suitable

for obtaining clearly expressed features in the global correlograms at periods of 15–50 s[30,32,62].

The data processing follows the steps described in previous studies[29,30]. These include removing the mean and linear trend, correcting the instrumental response to the velocity component, and resampling the waveforms to 10 samples per second. We apply temporal and spectral normalizations to enhance the expression of correlation features. The first normalization used the running-absolute-mean method in the time domain[63]. To further reduce the contribution of dominant frequencies, we normalize the waveforms' complex spectra using a spectral whitening operation[64,65]. The cross-correlation is computed in the frequency domain and inversely transformed back to the time domain. Cross-correlated waveforms are folded in lag time 0 to retain symmetric correlation functions. Finally, they are linearly stacked in 1° bins for inter-station distances over 0–180°. The correlograms of all events are linearly stacked and bandpass filtered between 15 and 50 s.

### Selection of event focal mechanism and station pair geometry
The selection of seismic source and receiver pairs used to construct the global correlogram is based on the recent understanding of the formation mechanism of correlation features. Earthquakes with normal or reverse faulting mechanisms efficiently radiate vertically-reverberating seismic energy through the Earth's center[29,33], and have been shown to enhance the overall expression of the correlograms[41]. In the late coda, the seismic energy still retains its direction of propagation, and the cross-correlation of earthquake records at two stations constructively contributes to the formation of correlation features if the station pair forms a great circle plane with the source. The cross-correlation of records at receiver pairs away from the same great circle plane leads to destructive interference[29,39].

Firstly, we categorize earthquakes into normal, thrust, and strike-slip faulting mechanisms on the ground of their cataloged dip and rake angles. Events with a dip angle larger than 70° and a range of rake angles between 70° and 110° are regarded as reverse faulting events, while ones having similar dips but rake angles between −110° and −70° are regarded as normal events. Only data from either normal or reverse faulting events are further processed. Secondly, we reduce the number of correlation pairs based on their relative locations with respect to the source. For each earthquake, only the cross-correlation of records at stations where the angular difference between their great circle plane with the sources is smaller than 20° are stacked. This selection is necessary to avoid destructive interference of seismic waves and enhance the quality of the features. Consequently, the improvement in the SNR of the correlogram after the selection of source mechanisms and station pairs are striking (Fig. 2b), and the observation of many correlation features becomes prominent.

In a recent study, Tkalčić and Phạm[36] used the coda correlation wavefield to probe the solidity of the IC by analyzing the I2-J, a correlation feature with the shape of a cusp that is observed at small inter-station distances. The I2-J has a prominent amplitude enhanced due to the spherical focusing effect[39], and its cusp is formed by a positive and a negative slowness leg that converges towards smaller inter-station distances. However, their correlogram was constructed by calculating the correlation functions of records of all events and receiver pairs available at IRIS DMC (Fig. 2a), which has been shown to result in destructive interference of seismic phases forming the correlation features. Such a procedure would render the detection of J-wave-sensitive correlation features, including the positive slowness leg of I2-J, under the observational threshold. Here, we overcome previous observational limitations and successfully enhance the expression of the correlation features in the global correlogram (Fig. 2b).

### Numerical full-waveform simulations
We perform several numerical simulations that served two purposes (i) examining the sensitivity of the correlation features to the background Earth velocity model and (ii) finding the best-fitting shear-wave speed in the IC through waveform modeling of the observations.

We use Yspec[43] to calculate the synthetic seismograms of the ten earthquakes that were used to successfully constrain the internal velocity structure of the Earth (CCREM[42]). Yspec is an efficient software for calculating full seismic wavefield in spherically symmetric Earth models using the direct radial integration method. The waveforms are calculated using PREM as the reference Earth velocity model. Apart from being computationally expensive, calculating the synthetic records of all the 139 normal and thrust seismic events and adding them to the correlogram stacks does not affect the overall quality of the synthetic correlograms nor change the timings of the correlation features in the synthetic scenario.

The moment tensor solution is obtained from the global centroid moment tensor catalog[61]. Ten-hour-long seismograms are calculated using the same source depth and event-receiver geometry configurations as in the real data. The processing and calculation of the correlograms and slant-stacks follow the same steps described for the real data. We slightly vary different parts of the PREM model for all numerical experiments to understand the sensitivities of several correlation features to uncertainties in representing the Earth's structures.

Because we analyze waveforms at periods of 15–50 s, the original Earth model PREM (established for 1 s) must be corrected for the dispersion effect. We use the central period of $T = 23.1$ s. In all instances, we refer to the variation of the Earth model PREM with respect to dispersion-corrected velocity models (see ref. 36 for more details on the frequency correction).

### Sensitivity diagnosis to the Earth's interiors
The first experiment aimed to diagnose correlation features that are sensitive to shear-wave speeds in the Earth's IC. We compute synthetic correlograms for a series of Earth models using PREM as a background model, but the IC shear-wave velocity profiles are perturbed from the original profiles from −6 to 6% (shown in Supplementary Movie 1). In Supplementary Movie 1, four correlation features are found highly sensitive to the J-wave speeds as they show strong timing variations according to the amount of shear-wave velocity perturbation.

To improve the visibility of the correlation feature, we calculate the phase-weighted stack of cross-correlated waveforms surrounding the features of interest (Table S1). The slant-stacking reveals the time and slowness of the seismic energy arriving at the receivers, indicating the features' geometrical slope in distance-time space (Figs. 5 and S1).

In a similar fashion, the second experiment aimed to diagnose the sensitivities of correlation features to uncertainties in the P-wave velocity profile in major layers of the Earth's interiors. The P-wave speed profiles in the mantle, outer core, and inner core are increased, kept unchanged, or decreased by 1% from the PREM, one change at a time, meanwhile keeping the rest of the Earth's velocity model unchanged. Figure 5 and S4 shows the comparison of slant-stacked synthetic correlograms with their observational counterparts. We can observe that the timings of all correlation features vary depending on uncertainties in the Earth velocity models (Fig. 5c–e). Interestingly, the I-J feature's timing is minimally sensitive to the subjective choice of background velocity models used for other parts of the Earth.

The insensitivity of I-J to variations in the mantle and outer core's compressional wave velocity profile can be understood intuitively based on the ray geometry of its constituents. I-J is formed by the time difference of seismic phases (e.g., PKiKP and PKJKP, or PKIKP and PKJKP), generally having the same segments in their ray paths in the mantle and outer core (P and K segments). Thus, the perturbation in the velocity profiles in those regions

affects their delay times identically so that the time variations are canceled out in the I-J feature's differential timing. The I-J insensitivity to variations in P-wave velocity in the IC can also be intuitively understood based on the ray geometry. The compressional constituent of I-J spends little time in the IC (note that the PKIKP only samples the very top of the IC, as shown in Fig. 1). Thus, time changes in the compressional phase constituent of I-J are subtle and do not significantly affect its overall travel times.

As for the remaining correlation features (I2-J, x1-J, and x2-J), their relative dependence on the background velocity model justifies the different estimates of shear-wave speed in the IC. Even though these features are sensitive to J waves, they are also strongly sensitive to the compressional velocities assumed in other layers of the Earth, including the IC. Thus, differently from the I-J feature, the I2-J, x1-J, and x2-J are contaminated by excursions of PREM from the actual Earth's interior seismic velocity at 15–50 s[42].

### Waveform modeling of I-J correlation feature

The third numerical simulation was designed to find the best-fitting velocity model of J waves based on the waveforms of I-J. We perturb the IC shear wave velocity from −6 to 6%, in steps of 0.5%, relative to PREM, and keep the rest of the Earth model unchanged. In each perturbation, we calculate the correlogram and then compute the phase-weighted slant-stack of I-J. We do the same for the observation. We use the Numpy correlation coefficient (CC) routine (Harris et al., 2020) as a statistical measure to establish the best-fitting model (Table S1), where CC = 1.0 indicates a positive relationship between the two waveforms (i.e., highest coherence). CC is calculated between the slant-stacked waveforms of observation and synthetics for a range of ray parameters (from 1.0 s/° to 2.0 s/°, every 0.1 s/°). The phase-weighted stacked waveforms are stable for any choice of slowness, as illustrated by the high CC value at −3.4% velocity reduction in Fig. 5b. This reinforces that the modeling of any waveform extracted from the slant stacks at 1.0 s/° to 2.0 s/° converges to the same best-fitting shear wave speed (Fig. 5a, b). For each shear-velocity perturbation, we model the ensemble of waveforms (one waveform for each ray parameter) to obtain the best-fitting J-wave speed.

The shear wave speed uncertainty is ±0.5%, equivalent to the steps used for perturbation in the IC shear-wave speed in the synthetic models. For comparison, we also perform waveform modeling based on I2-J, x1-J, and x2-J. Even though these features are affected by uncertainties in the background velocity model (Fig. S3, Table S1), they might be used in future studies to constrain seismic properties of the mantle and outer core. The timing and distance windows used to calculate the phase-weighted stacking of I-J and all other correlation features are outlined by the rectangles in Fig. S1a and documented in Table S1.

### Data availability

The seismograms and related metadata used in this study were downloaded from IRIS-DMC (available at https://ds.iris.edu/ds/nodes/dmc/data/types/waveform-data). The earthquake catalog is accessible from the GCMT database (available at https://www.globalcmt.org/CMTsearch.html).

### Code availability

The code for computing the cross-correlation of the late coda is available in ref. 29. The Yspec code used to generate the synthetic waveforms is referred to in ref. 43.

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

## Acknowledgements

The numerical simulations were undertaken with the assistance of resources and services from the National Computational Infrastructure (NCI) facility supported by the Australian Government. The processing of the seismograms and computation of the cross-correlograms were performed in the computing facilities at the Research School of Earth Sciences (RSES), The Australian National University. The authors would like to thank Dr. Sheng Wang for meaningful discussions that helped to solidify the methodological approach used in this work. The Australian Research Council supported this work through a Discovery Project (DP220102815). T.-S.P. acknowledges support from the Australian Research Council through a Discovery Early Career Research Award (DE230100025). T.C.L. acknowledges the ANU Ph.D. scholarship support through her degree.

## Author contributions

All authors contributed to the conceptualization of the study. T.C.L., X.M., and T.-S.P. carried out the data processing, numerical simulations, and analysis in consultation with H.T. T.C.L., and T.-S.P. wrote the first draft, which all other authors then improved. This work is part of T.C.L.'s Ph.D. research project under the supervision and guidance of H.T.

## Competing interests

The authors declare no competing interests.
