## [Peer Review File · Nature Communications]

An estimate of absolute shear-wave speed in the Earth's inner coreREVIEWERS' COMMENTS

Reviewer #1 (Remarks to the Author):

Shear wave velocity of the inner core is very important for understanding dynamic of the Earth's core, yet has not been well resolved yet. This manuscript presents nice study of PKJKP wave from earthquake coda interferometry and provides new constraints on shear velocity of the inner core. The authors found that the shear velocity is a few percent lower than standard Earth reference models, thus raising questions for further investigations. The manuscript is well written and the evidences are quite sound. I only have a few minor comments:

(1) about data selection. It seems that the I-J part of the correlograms before data selection is also quite good (time around 600s and distance around 120 degree).

(2) about synthetic data. A synthetic for the case of complete liquid inner core would be helpful for confirming the phase as PKJKP wave.

(3) Shear velocity of inner core may show lateral variation (Zeng and Ni,2013. Constraining shear wave velocity and density contrast at the inner core boundary with PKiKP/P amplitude ratio,Journal of Earth Science,DOI: 10.1007/s12583-013-0371-9). Thus, the shear velocity in this manuscript is an averaged results. The authors may discuss the effect of lateral variation and the raypath coverage. For example, much fewer stations are in southern hemisphere.

(4) about the physical mechanisms for lower Vs in the inner core. A recent paper by Wang et al (2021) is relevant.

Wang, Wenzhong; Li, Yunguo; Brodholt, John; Vočadlo, Lidunka; Walter, Michael J.; Wu, Zhongqing. "Strong shear softening induced by superionic hydrogen in Earth's inner core", Earth and Planetary Science Letters, 568, 117014.

Reviewer #2 (Remarks to the Author):

I have carefully reviewed this manuscript on the properties of Earth's inner core, and find it to address a topic of broad interest for the Earth and planetary sciences communities. The authors' conclusion that the shear velocity of the "solid" inner core is less than previously thought is important, as this observational result bears directly on the constitution and state of our planet's deep interior.

Also important is that the result is based on a new class of seismological observations that have broad applicability to other regions within our planet.

The analysis is clear, although it seems to assume a radial velocity profile for the mantle and core (if I understood correctly), and it would be nice to be reassured that the authors have thought about possible bias due to lateral heterogeneities. In particular, lateral heterogeneities within the IC might systematically affect the I-J results, perhaps even strengthening the authors' conclusions. This is a minor (and speculative) point that does not undermine the importance of the authors' primary conclusions, however.

Reviewer #1 (Remarks to the Author):

Shear wave velocity of the inner core is very important for understanding dynamic of the Earth's core, yet has not been well resolved yet. This manuscript presents nice study of PKJKP wave from earthquake coda interferometry and provides new constraints on shear velocity of the inner core. The authors found that the shear velocity is a few percent lower than standard Earth reference models, thus raising questions for further investigations. The manuscript is well written and the evidences are quite sound. I only have a few minor comments:

Thank you for your positive and thoughtful comments on our manuscript. Please find below our responses.

(1) about data selection. It seems that the I-J part of the correlograms before data selection is also quite good (time around 600s and distance around 120 degree).

We suspect you are referring to cKP-cP feature (see Pham et al., 2018, for the naming convention) instead of the I-J feature. cKP-cP, most sensitive to the outer core structure and insensitive to the IC, is a prominently expressed feature in observed correlograms; see Figure 2a in this paper and Figure 2 in Tkalčić and Pham (2018). On the other hand, I-J is a much weaker feature appearing near the tip of cKP-cP around 600s and 120 degrees, marked by an arrow in Figure 4a. The data selection procedure, suggested by Wang and Tkalčić (2022), proved crucial in improving the observability of I-J in real-data correlograms. Note that Tkalčić and Pham (2018) noticed the I-J feature in the synthetic correlogram but could not observe it back then without the data selection we performed in this study, so they resorted to I2-J.

(2) about synthetic data. A synthetic for the case of complete liquid inner core would be helpful for confirming the phase as PKJKP wave.

We acknowledge your comment. Our numerical experiment, introduced by Tkalčić and Pham (2018) to detect J-wave sensitive features in the coda correlation wavefield, aimed to achieve what you meant by the liquid IC test. In that experiment, the variations of J-wave speeds in the IC are incremental, from -6% to 6%, relative to PREM values by a 0.5% step. The change in J-wave speeds results in time variation, i.e., the vertical movement of four correlation features (Lines 140–144 in the main text). Because the IC is the centermost shell of the Earth, accounting for a small fraction of the Earth's volume, the response is evident for the sensitivity of the correlation features to J-wave speeds.

A liquid IC test, where $v_s = 0$ everywhere in the IC, is equivalent to an extreme case of our experiment, a -100% reduction from PREM values. Therefore, a synthetic test with a fully liquid IC is, we believe, unnecessary to demonstrate the sensitivity to J-wave speeds.

(3) Shear velocity of inner core may show lateral variation (Zeng and Ni, 2013. Constraining shear wave velocity and density contrast at the inner core boundary with PKiKP/P amplitude ratio, *Journal of Earth Science*, DOI: 10.1007/s12583-013-0371-9). Thus, the shear velocity in this manuscript is an averaged results. The authors may discuss the effect of lateral variation and the raypath coverage. For example, much fewer stations are in southern hemisphere.

Thank you, the work you suggested is now mentioned in Lines 267–268 of the main text.

As you noted, the estimate of the J-wave speed in this study is assumed to be globally averaged. Because of this assumption, the lateral distribution of earthquakes and stations on Earth's surface does not impact our result. It is indeed an interesting question for following-up research on how the current distribution of sampling coverage affects our understanding of more

complex S-wave structure in the IC. We added a new paragraph in the main text, Lines 319–329, to expand our thoughts.

(4) about the physical mechanisms for lower Vs in the inner core. A recent paper by Wang et al (2021) is relevant.

Wang, Wenzhong; Li, Yunguo; Brodholt, John; Vočadlo, Lidunka; Walter, Michael J.; Wu, Zhongqing. "Strong shear softening induced by superionic hydrogen in Earth's inner core", *Earth and Planetary Science Letters*, 568, 117014.

Thank you for suggesting this work, which is indeed relevant. We have added this reference to Figure 1b, its caption (ref 5 in line 85), and appropriate places elsewhere (lines 45, 101, 304).

References

- Phạm, T.-S., Tkalčić, H., Sambridge, M., & Kennett, B. L. N. (2018). Earth's Correlation Wavefield: Late Coda Correlation. *Geophysical Research Letters*, 45(7), 3035–3042. <https://doi.org/10.1002/2018GL077244>
- Tkalčić, H., & Phạm, T.-S. (2018). Shear properties of Earth's inner core constrained by a detection of J waves in global correlation wavefield. *Science*, 362(6412), 329–332. <https://doi.org/10.1126/science.aau7649>
- Wang, S., & Tkalčić, H. (2022). Scanning for planetary cores with single-receiver intersource correlations. *Nature Astronomy*, 1–8. <https://doi.org/10.1038/s41550-022-01796-8>

Reviewer #2 (Remarks to the Author):

I have carefully reviewed this manuscript on the properties of Earth's inner core, and find it to address a topic of broad interest for the Earth and planetary sciences communities. The authors' conclusion that the shear velocity of the "solid" inner core is less than previously thought is important, as this observational result bears directly on the constitution and state of our planet's deep interior.

Also important is that the result is based on a new class of seismological observations that have broad applicability to other regions within our planet.

The analysis is clear, although it seems to assume a radial velocity profile for the mantle and core (if I understood correctly), and it would be nice to be reassured that the authors have thought about possible bias due to lateral heterogeneities. In particular, lateral heterogeneities within the IC might systematically affect the I-J results, perhaps even strengthening the authors' conclusions. This is a minor (and speculative) point that does not undermine the importance of the authors' primary conclusions, however.

We appreciate your positive feedback and the thought-provoking comment on our manuscript. Indeed, given the lateral heterogeneity observed in the P-wave structure, the IC might have a similar, if not higher, lateral heterogeneity in its S-wave structure. However, being able to quantify the exact effect of the lateral heterogeneities on the timing and amplitudes of the I-J feature is beyond the scope of this work. This might require a further selection of event-receiver pairs and an even better understanding of the sensitivity kernel of the correlation wavefield, including its finite frequency kernel. To extend our thoughts on this comment, we added a new paragraph, Lines 319–329, in the main text.